# Occurrence of Ochratoxin A in Different Types of Cheese Offered for Sale in Italy

**DOI:** 10.3390/toxins13080540

**Published:** 2021-08-02

**Authors:** Alberto Altafini, Paola Roncada, Alessandro Guerrini, Gaetan Minkoumba Sonfack, Giorgio Fedrizzi, Elisabetta Caprai

**Affiliations:** 1Department of Veterinary Medical Sciences, University of Bologna, 40064 Ozzano dell’Emilia, Italy; alberto.altafini@unibo.it (A.A.); alessandro.guerrini5@unibo.it (A.G.); 2 Reparto Chimico Degli Alimenti, “Bruno Ubertini” Experimental Zooprophylactic Institute of Lombardy and Emilia Romagna, 40127 Bologna, Italy; g.minkoumbasonfack@izsler.it (G.M.S.); giorgio.fedrizzi@izsler.it (G.F.); elisabetta.caprai@izsler.it (E.C.)

**Keywords:** ochratoxin A, cheese, food safety, LC-MS/MS

## Abstract

The detection of Ochratoxin A (OTA) in the milk of ruminants occurs infrequently and at low levels, but its occurrence may be higher in dairy products such as cheese. The aim of this study was to investigate the presence of OTA in cheeses purchased in the metropolitan city of Bologna (Italy) and the surrounding area. For the analysis, a LC-MS/MS method with a limit of quantification (LOQ) of 1 µg/kg was used. OTA was detected in seven out of 51 samples of grated hard cheese (concentration range 1.3–22.4 µg/kg), while it was not found in the 33 cheeses of other types which were also analysed. These data show a low risk of OTA contamination for almost all types of cheese analysed. To improve the safety of cheese marketed in grated form, more regulations on cheese rind, which is the part most susceptible to OTA-producing moulds, should be implemented or, alternatively, producers should consider not using the rind as row material for grated cheese. It would be interesting to continue these investigations particularly on grated hard cheeses to have more data to update the risk assessment of OTA in cheese, as also suggested by EFSA in its 2020 scientific opinion on OTA.

## 1. Introduction

The whole world is facing a large crisis due to the coronavirus pandemic (COVID-19) which began in 2019 in China, from where it rapidly spread all over the world. The pandemic affected food industries around the globe, in sectors such as agriculture, food supplies and animal production [1], causing an adverse impact on poverty, food security, and diets [1,2]. The concept of food security includes four components: availability, access, utilisation (quality and safety), and stability; and five elements that are recognised to be critical in determining food environment: proximity, convenience, availability, affordability, and quality [3].

In order to combat the COVID crisis, food companies confirmed the implementation of more restrictive hygiene procedures as well as the additional purchases of PPE. Despite all of these challenges, available data seemed to indicate that food safety had not been compromised [4]. In this context, we believe it is important to emphasise that, in addition to all the new measures taken to reduce the risk of coronavirus contamination in food, regulations in the food industry, already in place before the pandemic, should not be overlooked. Among these, those concerning environmental contaminants, including mycotoxins, are of primary interest. The present study concerns the detection of a well-known mycotoxin (Ochratoxin A) in cheeses.

Cheese is one of the oldest foods in human history. It was born as a “natural preservation” of milk. In fact, it was the easiest way to keep a liquid and otherwise perishable raw material in a solid form [5]. According to the Italian legislative definition, cheese is the product that is obtained from whole, partially or totally skimmed milk, or from cream, as a result of coagulation with acid treatment, or rennet, which use of ferments and salt [6].

The wide variety of cheeses makes a single general classification of them impossible. They can be subdivided on the basis of different characteristics such as the origin, the ripening time, the temperature of the curd, the percentage of fat or water, and the consistency of the paste. Other elements are the breed of the animal and the treatment of the milk (raw, pasteurised, whole, or skimmed) [7,8,9,10]. 

Each cheese could belong to more than one reference group. In order to avoid this problem, the main reference is generally the fundamental characteristic of the cheese in question [9]. In Italy, there is a large variety of typical cheeses that are made in a specific way, according to gastronomic heritage. They are naturally processed, and distinguished because of their sensory properties and association with a certain local area or region [5,8]. Several consortiums have been established for the protection of these products and to obtain the recognition of them as a PDO (Protected Designation of Origin), PGI (Protected Geographical Indication) or TSG (Traditional Specialties Guaranteed) product [9,11]. Furthermore, a large number of cheeses (more than 450) are included in a national list of traditional agro-alimentary products (PAT), which is updated by the Italian Ministry for Agricultural, Food and Forestry Policies in collaboration with the appropriate regions every year [9,12].

Cheese is a staple food for many populations in Europe and around the world [13], and is one of the excellences of Italy. This food sector is extremely important for the Italian economy. According to 2019 data, at EU level, Italy is the third largest cheese producer producing over 1.2 million tonnes per year, after Germany (2.3 million tonnes) and France (1.9 million tonnes). Furthermore, Italy is the largest producer of PDO cheeses, while the EU is the world’s largest exporter of cheese (over 900,000 tonnes in 2020) [14].

Mycotoxins are toxins produced by some fungi such as mushrooms, yeasts, and moulds. Some of them are able to produce toxins only in particular conditions while others are not toxigenic [15]. Moulds are generally considered as undesirable in food, but that is not always the case. In fact, in several meat and cheese products, the growth of spontaneous microflora, characteristic of the geographical environment from which the food originates, plays an important role, especially in the typical productions, in the development of characteristic flavours and aromas. Furthermore, moulds can be intentionally added to food products to induce the formation of specific flavours [16]. Classic examples are blue cheeses such as Roquefort and Gorgonzola, characterised by an extensively spotted or veined structure due to blue or blue–green moulds [17], and cheeses ripened with surface mould (for example, Camembert and Brie) [18,19]. Among meat products, typical examples are ham, speck, Bündnerfleisch, coppa and dry-cured sausages. 

The shift in production of these foods from small local producers to large-scale factories and the need to assure a high level of safety towards mycotoxin production has resulted in an increasing use of commercial mould starter cultures [20]. These cultures may also act as biocontrol agents even though they had not been selected for their properties to inhibit unwanted microorganisms [21]. Protective cultures inoculated during cheesemaking may actively compete with potentially toxigenic fungal contaminants and hinder their spread through the production of inhibitory compounds [22]. A study carried out on white mould cheese has shown that two secondary starters, *Penicillium camemberti* and *Geotruchum candidum* can have an inhibitory effect on toxigenic mould contaminants isolated from either the dairy environment or directly from the cheese [23]. Delgado et al. reported that some moulds belonging to the genera *Aspergillus*, *Fusarium*, *Monascus*, *Neosartorya*, and *Penicillium* produced various peptides and proteins with antifungal properties, and that such moulds could be a useful tool to control mycotoxigenic fungi [21]. However, even mould species that contribute positively to the maturation process and to the development of the typical organoleptic characteristics of certain type of cheeses, in cases where they grow on the wrong cheeses, they are considered contaminants that cause great economic losses for food manufacturers [24]. Mycotoxins reported to contaminate cheese include: sterigmatocystin, penicillic acid, patulin, PR toxin, roquefortine, citrinin, cyclopiazonic acid, mycophenolic acid, aflatoxin, isofumigaclavine A, ochratoxin A, and penitrem A. Among all of these mycotoxins, the most hazardous are aflatoxin M1 and ochratoxin A [22,25]. The presence of these substances in dairy products can be due to indirect contamination of the milk used for cheese production, which comes from animals fed with contaminated feed, or can be due to direct contamination of the cheese caused by the growth of mycotoxigenic moulds [22,25,26,27,28]. In the cheese production plants, air contaminated with mould spores represents the most major source of mould contamination [29,30,31,32].

Ochratoxin A (OTA), the mycotoxin on which the present study is focused, is the most prevalent and most toxic of the ochratoxins. This group of structurally similar molecules are poisonous secondary metabolites produced by fungi belonging to the genera *Aspergillus* and *Penicillium* [15,33,34]. OTA has been shown to have nephrotoxic, hepatotoxic, carcinogenic, teratogenic, genotoxic and immunotoxic properties, as well as other hazardous properties [35,36,37]. In 1993, OTA was classified by the International Agency for Research on Cancer as a possible carcinogen (Group 2B) to humans [38].

Several studies also report that OTA may be involved in the pathogenesis of different forms of human nephropathies such as the Balkan Endemic Nephropathy (BEN) [39,40,41] and the Chronic Interstitial Nephropathy (CIN) [42,43], but the true role of this toxin in the etiology of these diseases is still debated [44,45,46,47]. OTA has been detected in a wide variety of agricultural commodities, including beans, cocoa, corn, dried fruits, grapes, raisins, grape juice, wine, beer, coffee, rice, spices, wheat, barley, oats, and rye, and can also contaminate livestock products such as meat, milk and their derived products [48,49,50]. Recent studies have also shown OTA to be present in alfalfa, food colours, spices and even in bottled water [51]. The food categories representing the main contributors of human exposure to OTA are preserved meats, cheeses, grains, and grain-based products [52]. OTA is frequently present in various feedstuffs, in particular cereals (maize, barley, wheat, oats, rice) and cereal by-products [36,53].

Among farmed animals, pigs are particularly susceptible to the adverse health effects associated with OTA exposure, and the toxin can accumulate in the tissues of such animals [54,55,56]. Pigs are also one of the most frequently exposed species due to their mainly cereal-based diet [57]. Compared to monogastric animals, ruminants are less sensitive to OTA exposure because the protozoan fraction of rumen fluid is capable of enzymatic degradation into ochratoxin-α (OTα) and other less toxic metabolites [58]. However, the rumen bypass of a certain percentage of intact, undegraded OTA needs to be taken into account, and this rate could range from 5 to 62% [59]. Since in these animals the bioavailability of OTA is low to begin with, the chance of OTA transferring to milk is minimal. Thus, detection of the toxin in milk generally occurs only in cases of high concentrations of OTA in animal feed [60,61].

Guidance values for OTA have been established at a European level for feed materials, in particular cereals and cereal products (0.25 mg/kg), and complementary and complete feedstuffs for pigs (0.05 mg/kg) and poultry (0.1 mg/kg) [62]. The European Union adopted maximum permissible limits for OTA in several foodstuffs such as cereals, dried vine fruits, coffee, wine, grape juice, foods for infants and young children, spices, liquorice, and wheat gluten, and range from 0.5 to 80 μg/kg [63]. In contrast, there are no regulations in the European Union for mycotoxins in milk, except for aflatoxin M1 and the hydroxylated derivative of aflatoxin B1, which occur in the milk of lactating animals [64]. Among the EU member states, only Slovakia has set a maximum level for OTA in milk (5 µg/kg) [65,66]. 

Even though the detection of OTA in the milk of ruminants occurs infrequently and at low levels, OTA occurrence may be higher in dairy products such as cheese [52]. However, relatively few data are available about the presence of OTA in cheese. The European Food Safety Authority (EFSA) Panel on Contaminants in the Food Chain (CONTAM Panel), in its 2020 scientific opinion on ochratoxin A (OTA) in food, reported that more data on OTA in cheese paste in comparison to cheese rinds are needed [52]. 

The aim of the present study was to investigate the presence of ochratoxin A in various type of cheese purchased from local markets, supermarkets, and specialised shops, mostly located in the metropolitan city of Bologna (Emilia Romagna, Northern Italy) and the surrounding area. Priority was given to the purchase of Italian products produced in the regions Emilia Romagna and Lombardy. Furthermore, a significant part of the samples were packages of ready-to-use grated cheese, which are increasingly popular because of their convenience and packaging that allows proper storage in the refrigerator. For the analysis, a sample preparation procedure with immunoaffinity columns (IACs) and a liquid chromatography-tandem mass spectrometry system (LC-MS/MS), were used.

## 2. Results

### 2.1. Method Validation

In all regressions, linearity was satisfactorily higher than the considered range (0.1–20 µg/L). The coefficient of determination (R^2^) was always >0.99 showing a good agreement between peak area and OTA concentration. Assay interference was evaluated by analyzing cheese samples with no detectable amount of OTA and spiked samples (from 1 to 40 µg/kg). No interfering peaks were observed in the spiked samples and no significant peaks were found at the retention time of OTA in the blank samples, showing a satisfactory specificity. The retention time of OTA in naturally contaminated samples and standard solutions was about 2.7 min while the total run time was 3.5 min. Figure 1 shows a chromatogram of a naturally contaminated cheese at a 5 µg/kg level for the mass transitions monitored for U-[^13^C_20_]-OTA (internal standard [IS]) and OTA.

Recovery was assessed at three spike levels (six replicates for 3 days), and the average recovery percentages ranged from 80.17% to 84.43%. The results are shown in Table 1, and the overall average recovery was 82.9%. 

The repeatability and reproducibility tests were based on intraday and inter-day measurements. The relative standard deviations (RSDs) of quantification results ranged from 7.22 to 10.54% and from 9.15 to 13.26%, respectively (Table 2).

These results comply with the performance criteria fixed by Regulation (EC) 401/2006 of the Commission of the European Communities [67].

The LOQ and the LOD values obtained were 1 and 0.5 µg/L, respectively. These values underline the good level of sensitivity of the analytical method used, for example, up to the maximum level of OTA in milk established by Slovakia (5 µg/kg), the only country of the European Union who has set a limit for OTA in such this matrix [65,66]. Overall, the validation results show that the methodology applied in this study performed well in quantitating OTA in the products analysed.

### 2.2. Occurrence of Ochratoxin A in Cheese Samples

In the present study, a total of 84 cheeses of various type were analysed for the presence of OTA. Firstly, 73 cheeses purchased at the start of the study were tested; 69 of them were below LOQ, while four samples showed a concentration level of OTA ranging from 1.3 to 7.5 µg/kg. In particular, all of these positives were samples of grated hard cheese made from cow’s milk. They were all Italian grana-type cheeses produced in the regions Emilia Romagna (*n* = 2), Lombardy and Puglia. The first two were 30-months-matured cheeses with designations of origin (PDO), while the others were products derived from a mix of different hard cheeses. Thus, the percentage of positives out of the total number of cheeses analysed was 5.5%, while, when considering only the category of grated hard cheeses (*n* = 40), this percentage increased to 10%. Furthermore, among all the samples examined, six were grana-type cheeses packaged in whole portions, and five of them also included the rind. In these samples, after analyses of the cheese paste, further investigations were also carried out on the rind but OTA was not detected on this part. Following these findings, an additional sampling focused on grated hard cheeses was carried out. Eleven further cheese samples of this type were collected and analysed. OTA was found in three samples at a concentration level ranging from 3.2 to 22.4 µg/kg. These positives were PDO grana-type cheeses produced in Emilia Romagna, one of them matured for at least 30 months and the others for at least 12 months. Thus, the percentage of positives found in the additional sampling was 27.3%, while, by considering the total number of cheeses found contaminated by OTA (*n* = 7), the percentage of positives was 13.7% and 8.3% with respect to the total number of grated hard cheeses (*n* = 51) and the total number of cheeses of various type collected for the present survey (*n* = 84). Cheese samples and OTA concentrations are reported in Table 3.

## 3. Discussion

If we look at the cheeses analysed as a whole, based on the percentage of positives, we are led to conclude that, in general, OTA risk in this type of food is very low, but for a better interpretation of the results, it is necessary to extrapolate the grated hard cheese category and make a separate analysis of the data. The percentage of positive among these samples was not at all negligible, especially in the additional sampling. As for the concentration of OTA detected, while in two samples it was slightly above the LOQ and in the other two samples it was higher but ≤5 µg/kg, this concentration was above 7 µg/kg in the other two samples. This last contamination level is not so low if we consider the maximum permitted level of OTA in milk (5 µg/kg) established by Slovakia is taken as a reference, as the European Commission has not yet set a limit for OTA in cheese. Finally, an OTA level equal to 22.4 µg/kg found in one sample shows that a fairly high contamination by OTA is possible in this type of product. Thus, among the products sampled in this survey, the grated hard cheese category is the one at the highest risk of this kind of contamination. As previously reported, in this food product, a contamination due to the presence of OTA in the milk used for cheese production is rather unlikely. In fact, indirect contamination of the milk from animals fed with contaminated feed can happen only in the case of very high concentrations of OTA in animal feed, due to the low carryover of this toxin in ruminants [51,52]. In contrast to AFM1, few investigations have been carried out on the presence of OTA in animal milk. 

Valenta and Goll [68] reported no positive samples after an analysis to detect the presence of OTA in 121 regional milk samples from Germany. In a study carried out in Norway by Skaug [69], samples of organic and conventional cow’s milk, and cow’s-milk-based infant formulas were analysed for the occurrence of OTA. Six out of forty conventional milk samples (range 11–58 ng/L) and 5 out of 47 organic milk samples (range 15–28 ng/L) were found to be positive, while the mycotoxin was not detected in any of the 20 infant formula samples. A survey on the presence of AFM1 and OTA in samples of raw bulk milk was conducted in France in 2003 and OTA was found in 3 out of 264 samples at low levels, in a range from 5 to 8 ng/L [53]. Pattono et al. [70] analysed 63 samples of organic and 20 samples of conventional milk and OTA was detected in only three samples of organic milk with amounts comprised between 0.07 and 0.11 µg/kg. 

Higher percentages of positive samples were found in a survey carried out in Turkey in 2017. OTA was detected at concentration levels ranging from 10 to 270 ng/L in 37 out of 40 cow’s milk samples collected from milk collection tanks [71]. A subsequent study carried out in Turkey on 105 milk samples (35 raw, 35 pasteurized, and 35 UHT) showed an average level of an OTA concentration of 119 ± 19 ng/L [66]. These last two surveys both showed slightly higher levels of OTA than previous studies, and one of them showed the highest percentage of positives. Such results could also be due to the analytical method applied (ELISA test) that sometimes leads to overestimations of the number of positives.

Direct contamination of the milk used for cheese production also rarely occurs because such milk is normally pasteurised or heat-treated and mycotoxigenic mould spores are generally not heat-resistant [29,30,31,72]. Therefore, the most probable hypothesis is that the presence of OTA is due to the growth of ochratoxigenic moulds of environmental origin on the surfaces of cheeses. In the specific case of the grated hard cheeses, a certain aliquot of the product can be derived from the cheese rind, even in the case of PDO cheeses. For example, according to the specifications for Grana Padano and Parmigiano Reggiano cheese, the packages of grated cheese may contain up to 18% of product from the rind [73,74]. Furthermore, all of the grated cheeses examined in the present study were matured products. In the case of the positives, 3 samples were 30 months matured, 2 samples 12 months matured, while 2 samples were a mix of cheeses matured for periods ranging from 4 to 12 months. It is clear that if contamination occurs in the maturing premises, the maturing time could also play a certain role in the growth of ochratoxigenic moulds. Conversely, in the case of fresh cheeses, this kind of contamination normally does not occur. The maturing time of a few days is insufficient for the growth of contaminating filamentous fungi [22]. However, it should be noted that the six grana-type cheeses packaged in whole pieces that were part of the sampling, were matured cheeses (12 months) but, unlike their grated counterparts, they were all negative for the presence of OTA, including the respective rinds, which were analysed separately.

The findings of the present survey corroborate a previous study carried out in Italy by Biancardi et al. [75] in which 40 samples of commercial grated hard cheese were analysed and OTA was detected in six samples at concentrations equal to 1.62, 2.03, 5.28, 12.47, 14.15, and 54.07 µg/kg, respectively. 

In a survey which aimed to provide information on the mycobiota associated with Italian grana cheese, 18 cheese samples were collected and analysed for the presence of citrinin (CIT) and OTA. This latter mycotoxin was detected in all samples at concentrations ranging from 1 to 1432 μg/kg (mean 183 µg/kg). According to the authors, the high levels of contamination found can be explained by the fact that the aliquots of cheese for analysis were taken exclusively from the rind, which is the part of the cheese most exposed to contamination in the presence of ochratoxigenic moulds of environmental origin [76]. It seems clear that in many cases cheese rinds act as a protective system against external contamination during ripening. In a study which aimed to identify the fungal colonisation of ripening cave cheese (a traditional Italian cheese) and relate it to possible mycotoxigenic risk, OTA was found on the rind of 8 out of 22 samples in the range 0.2–317 µg/kg. In addition, in order to assess the ability of OTA to diffuse from the rind into the interior of the cheese, each sample was analysed at three different depth levels (the rind, the middle, and the central part), but OTA was not detected in any of the deeper parts [64].

However, removing the rind of the cheese before consumption, while good practice, does not prevent the consumer from the risk of OTA intake from cheese. In fact, in some cases OTA can cross the rind and contaminate the inner paste of the cheese, as shown in the survey carried out by Pattono et al. [72] which focused on the control of 32 traditional handmade semi-hard cheeses. OTA was detected both in the rind and in the inner part of six samples, with amounts ranging from 1 to 262 mg/kg and from 18 to 146 mg/kg, respectively.

A study carried out in Egypt by Younis et al. [77] examined the presence of OTA in 40 dairy products equally divided among row milk, milk power, Roomy cheese (hard cheese) and Kariesh cheese (soft cheese). The results showed that 80% of the samples were contaminated at concentrations ranging from 0.34 to 13 µg/kg (mean 5.134 ± 1.822 µg/kg). The highest incidence of OTA was found in Roomy cheese (concentration range 3–4.8 µg/kg), and was slightly higher than in Kariesh cheese (90% vs. 80%). According to the authors, this could have been due to the higher exposure of the matured cheeses to environmental ochratoxigenic moulds in the maturing premises. In contaminated fresh cheeses, if correctly stored, it is more likely that the occurrence of OTA may be linked to the milk used for cheese production. In fact, the short maturing time of these products is insufficient for the growth of contaminating moulds [22]. In the cited study [77], the finding of a high percentage of contaminated milk samples among those analysed (70% of fresh milk and 80% of powdered milk) and the absence of OTA-producing moulds on the surface of cheeses in 35% of the cheese products examined could support the above statement.

In case of detection of OTA in a cheese, it is always important and useful to be able to identify the origin of the contamination to determine which are the most critical points in the food production chain. Unfortunately, in many cases it is difficult to retrace the production chain to make the necessary checks. However, even from the analysis of the finished product, it is possible to find useful information to establish if the contamination is of environmental origin or due to the raw material. For example, one can observe the distribution of the toxin in the matrix (uniformly diffused, spot-like, and with a variable concentration from the outside to the centre of the cheese wheel), the presence of ochratoxigenic moulds, as well as the variation of the toxin content over time.

Zhang et al. [78] evaluated OTA concentration changes during a 2-week storage at 4 °C of two contaminated aged cheese samples, one of which had a mould stain. In the latter, a steady increase in the initial toxin concentration was found, with higher OTA levels in the area near the mould stain. These data indicate the presence of live OTA producing moulds in the cheese. In contrast, in the other sample, the toxin concentration remained constant, likely because there were inactive ochratoxigenic moulds. Furthermore, in this last case, the toxin was uniformly distributed in the cheese and this meant that the toxin was already present in the milk used to make the cheese. 

Finally, blue cheeses deserve a special consideration. This category of cheese accounted for a significant percentage (23.8%) of the total samples collected. It was interesting to assess whether these cheeses, characterised by an extensively spotted or veined structure due to blue or blue–green mould, could be at risk of OTA contamination. Furthermore, scientific literature reports only one survey of this type, which was carried out by Dall’Asta et al. [17]. The authors examined the presence of OTA in four different types of blue cheeses for a total of 92 samples: Gorgonzola cheese (*n* = 54), Blue cheese (*n* = 20), Roquefort cheese (*n* = 14), and Bergader cheese (*n* = 4). OTA was detected in 23 Gorgonzola samples (concentration range 0.2–3 µg/kg), and seven Roquefort samples (concentration range 0.1–1.4 µg/kg). The study also showed that the contamination was linked to the moulded spots of the cheese and therefore it was not due to raw material. Furthermore, additional tests carried out to evaluate any change in OTA concentration during the shelf life showed that the OTA level increased in contaminated samples but not in blank samples. These results suggested the presence of active OTA-producing moulds in contaminated cheeses. Nonetheless, the study did not establish whether OTA was produced by the industrial *Penicillium* strains used as starter culture or by an accidental concomitant microorganism.

For the present survey, in order to increase the number of types of blue cheeses to be tested, some typical cheeses from foreign countries (Germany, France and England) were also collected. However, distinct from the aforementioned study, the blue cheese samples analysed were all negative for the presence of OTA and further investigation on this type of cheese were not carried out.

## 4. Conclusions

In conclusion, the data of the present study generally seem to indicate a low risk for OTA contamination for almost all types of cheeses examined. However, among grated hard cheeses, the percentage of positive samples that were found is not at all negligible. While we are aware of the limited number of samples analysed, the fact that our data are in line with the similar survey conducted by Biancardi et al. [75], gives more weight to our results. To improve the safety of cheese marketed in grated form, more regulations focused on cheese rind, which is the part most susceptible to OTA-producing moulds, should be implemented or otherwise, producers should consider not using the rind as row material for grated cheese, even though such a measure would affect the final price of the product. Moreover, consumers of this type of food also include vulnerable sectors of the population, such as young children and the elderly, for which even greater caution is needed. Considering the lack of available data on OTA contamination in cheeses and the results of the present study, it would be interesting to continue these investigations particularly on grated hard cheeses. Such surveys may be useful to create more data to help update the risk assessment of OTA in cheese, as recommended by EFSA in 2020 [52].

## 5. Materials and Methods 

### 5.1. Samples

For this study, a total of 84 cheeses of various type were collected between January and April 2021. The cheeses were purchased mainly from local markets, supermarkets, and specialised shops, mostly located in the metropolitan city of Bologna (Emilia Romagna, north Italy) and the surrounding area. Priority was given to the purchase of Italian cheeses produced in the regions of Emilia Romagna, Lombardy and Piedmont. Each cheese was numbered and registered in an appropriate product data sheet. Furthermore, cheese packages with their ingredient labels were photographed and the photo files stored in order to have all available information about each cheese. Firstly, 73 cheeses were sampled. The main types were grated hard cheese (*n* = 40), and blue-veined cheese (*n* = 20). The other types were hard cheese (*n* = 6), soft cheese (*n* = 3), semi-hard cheese (*n* = 1), soft-ripened cheese (*n* = 1), spicy cheese (*n* = 1), and spun paste cheese (*n* = 1). Furthermore, most of the cheeses were from cow’s milk (*n* = 64), while the others were from ewe’s milk (*n* = 5) and goat’s milk (*n* = 4). For the analysis, in the case of hard cheeses with rinds, portions of rind and paste, collected separately, were sampled. Precautions were taken to avoid cross-contamination between the different parts of the cheeses. Portions that included white areas and areas with moulded spots were sampled in blue cheeses. All the cheeses were stored at −20 °C until analysis. 

After these 73 samples were tested for the presence of OTA, 11 additional grated hard cheeses were collected for the further investigation of this particular type of product, which was found to be the most contaminated by OTA.

### 5.2. Chemicals and Reagents 

The OTA standard and the U-[^13^C_20_]-OTA standard (internal standard, IS) used to prepare standard solutions for the validation of the analytical method were purchased from Sigma Aldrich (St Louis, MO, USA) and Or Sell (Limidi di Soliera (MO), Italy), respectively. Acetonitrile (LC-MS grade) and formic acid (LC-MS grade), methanol (LC-MS grade), n-hexane (analytical grade), and acetic acid (glacial) were purchased from Carlo Erba Reagents (Cornaredo, MI, Italy), Honeywell (Charlotte, NC, USA), Panreac Applichem (Barcellona, Spain), and Sigma Aldrich (St Louis, MO, USA), respectively. Ultrapure water used throughout the experiments was produced by an Evoqua Water Technologies system (Pittsburgh, PA, USA). Ochraprep^®^ immunoaffinity columns from R-Biopharm AG (Darmstadt, Germany) were used for samples purification.

### 5.3. Chromatographic Apparatus

Detection and quantification was conducted by LC-MS/MS on an Acquity UPLC (Waters, Milford, MA, USA) coupled to a Quattro Premier Xe triple-quadrupole mass spectrometer with electrospray ionisation source (Micromass, Manchester, UK). Based on the structural properties of the analytes, the positive ionisation modes (ESI+) were applied. The parameters were as follows: cone voltage, 25 V; capillary voltage, 1.0 kV; source temperature, 120 °C; and desolvation temperature 450 °C. Mass Lynx TM 4.1 SCN 805 software (Micromass, Manchester, UK) was used to control the instruments and process the data. The data acquisition was used in multiple reactions monitoring (MRM) mode. The analytical column was an Acquity UPLC BEH C18 1.7 µm 2.1 × 100 mm (Waters, Milford, MA, USA). Chromatographic separation was achieved in gradient elution mode. The mobile phase consisted of ultrapure water containing 0.2% formic acid/acetonitrile 95:5 (*v*/*v*) (eluent A) and acetonitrile containing 0.2% formic acid (eluent B). The gradient program started with 98% A and 2% B, reaching 40% A in 2 min with linear increase; then returned to 98% A in 0.5 min with a re-equilibration time of 1 min, giving a total run time of 3.5 min. The flow rate was 0.45 mL/min, while the sample injection volume was 10 µL. The ion transitions and mass parameters monitored for OTA and U-[^13^C_20_]-OTA, are reported in Table 4.

### 5.4. Sample Extract Preparation

The extraction of OTA from cheese was based on a method reported in a previous study [75] with some modifications, while the sample clean-up procedure was set up on the basis of the indications given in the instruction manual supplied with the IACs. Ten grams of minced sample were weighed in a suitable centrifuge tube, and 10 µL of an U-[^13^C_20_]-OTA standard solution 1 µg/mL were added as an internal standard (IS) (resulting in a final concentration in the sample of 1 µg/kg). After the addition of 50 mL of acetonitrile–water solution (84:16 *v*/*v*), the mixture was mixed for 60 min on a horizontal shaker and then passed through filter paper. A 4 mL aliquot of the upper phase was transferred into a centrifuge tube and 4 mL of hexane was added. After vortexing, the mixture was centrifuged at 3000 rpm for 5 min at ambient temperature and the hexanic upper phase was discharged. The defatting step with 4 mL of hexane was then repeated a second time. Finally, 3 mL of the extraction solution (equivalent to 0.6 g sample) previously diluted with 44 mL of PBS buffer (pH 7.4) was loaded onto an Ochraprep^®^ IAC.

The extract of each sample was then passed through the IAC at a flow rate of 1 drop/s. After a washing step with 20 mL of ultrapure water, the column was dried for several seconds using draw vacuum. OTA was eluted in a glass tube with 2 mL of methanol acidified, with 2% acetic acid. The eluate collected was reduced to dryness under a gentle stream of nitrogen at 45 °C. The residue was redissolved in 500 µL of methanol acidified with 2% acetic acid/water (1:1 *v*/*v*). After vortexing, the eluate was transferred in a glass vial before LC-MS/MS analysis.

### 5.5. Quantification

For the quantification of OTA in cheese, calibration curves were created by first preparing a set of OTA standard solutions in solvents at different concentrations and a constant amount of IS was added to all standards. After LC-MS/MS analysis, the curves were constructed based on the peak area ratio between OTA and U-[^13^C_20_]-OTA (IS) of each standard solution analysed. The OTA contents of the samples were calculated by extrapolating the peak-area ratio to the calibration curve.

### 5.6. Performance Evaluation

OTA solutions in solvent, blanks and spiked samples were used to check the performance of the adopted analytical method. Linearity was evaluated by analyzing OTA solutions in solvents at concentrations of 0.1, 0.25, 0.5, 1, 2.5, 5, 10 and 20 µg/kg, and matrix-matched OTA solutions at concentrations of 1, 5, 10, 20, and 40 µg/kg. The respective calibration curves (in solvent and in matrix) were generated and the coefficients of determination (R^2^) calculated. The linearity of the calibration curves was considered satisfactory if R^2^ > 0.99. Specificity was proved using 20 blank samples, which were analysed and evaluated for interference. The calculation of recovery was made by comparing the peak area ratio between OTA and U-[^13^C_20_]-OTA in spiked samples and the peak area ratio between OTA and U-[^13^C_20_]-OTA of pure standard solutions at the same concentration levels. The repeatability was calculated as the relative standard deviation (RSD%) of results obtained after fortifying six blank samples at three concentration levels (1, 10, 20 μg/kg) for a total of 18 determinations. The spiked samples were prepared and analysed with the same instruments, on the same day, and by the same operators. The laboratory reproducibility was evaluated by preparing and analyzing six blank samples fortified at three concentration levels (1, 10, 20 μg/kg). The procedure was carried out on three different days by different operators using the same chromatographic apparatus. The RSD% of the replicate measurements (54 determinations in total) was calculated. The limit of quantification (LOQ) and the limit of detection (LOD) were calculated on the basis of a signal-to-noise ratio at the OTA retention time of 10:1 and 3:1, respectively.

## Figures and Tables

**Figure 1 toxins-13-00540-f001:**
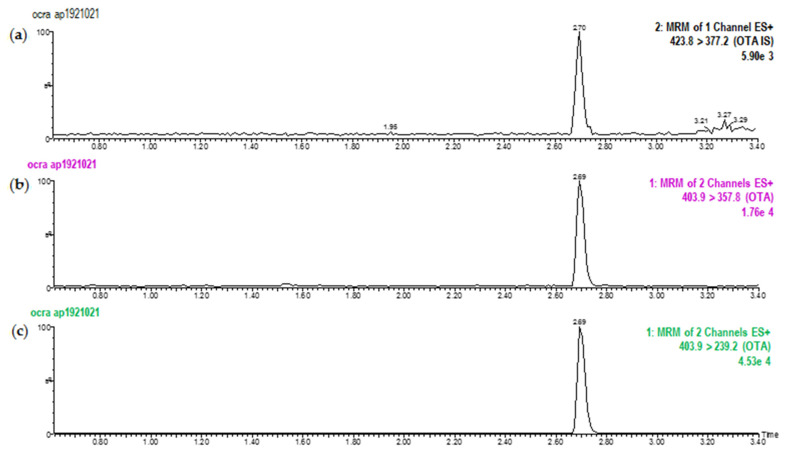
Chromatograms obtained after LC-MS/MS analysis of a naturally contaminated cheese at a 5 µg/kg level for the following mass transitions (**a**) m/z = 423.8 > 377.2 (IS), (**b**) m/z = 403.9 > 357.8, and (**c**) m/z = 403.9 > 239.2.

**Table 1 toxins-13-00540-t001:** Recovery data of the method for analysis of OTA in cheese samples spiked at 3 concentration levels.

	OTA Spiking Level (µg/kg)			M ^2^
	1	10	20	
Recovery (%) ^1^	84.43	80.17	84.27	82.9

^1^ Average of 18 replicates for each concentration; ^2^ average recoveries of the 3 spiking levels.

**Table 2 toxins-13-00540-t002:** Results of repeatability and reproducibility tests (expressed as RSD%) calculated for the analysis of OTA in cheese samples.

OTA Spiking Level (µg/kg)	Repeatability			Reproducibility		
	Mean (µg/kg)	SE ^1^ (µg/kg)	RSD ^2^ (%)	Mean (µg/kg)	SE ^1^ (µg/kg)	RSD ^2^ (%)
1	0.88	0.04	10.54	0.84	0.03	13.26
10	8.14	0.33	9.82	8.02	0.20	10.34
20	17.69	0.52	7.22	16.85	0.36	9.15

^1^ Standard error; ^2^ Relative standard deviation.

**Table 3 toxins-13-00540-t003:** Cheese samples and OTA concentrations.

No.	Type of Cheese	Milk Origin	Place of Production ^1^	OTA (µg/kg)	No.	Type of Cheese	Milk Origin	Place of Production ^1^	OTA (µg/kg)
1	Soft	Goat	Lombardy	<LOQ	43	Grated hard	Cow	Piedmont	<LOQ
2	Grated hard	Ewe	Sardinia	<LOQ	44	Grated hard	Ewe	Emilia Romagna	<LOQ
3	Grated hard	Cow	Emilia Romagna	<LOQ	45	Grated hard	Cow	Lombardy	<LOQ
4	Blue	Cow	Piedmont	<LOQ	46	Grated hard	Cow	Lombardy	<LOQ
5	Blue	Cow	Piedmont	<LOQ	47	Blue	Cow	Germany	<LOQ
6	Grated hard	Cow	Emilia Romagna	<LOQ	48	Blue	Cow	Piedmont	<LOQ
7	Grated hard	Cow	Veneto	<LOQ	49	Blue	Cow	Germany	<LOQ
8	Blue	Cow	Piedmont	<LOQ	50	Grated hard	Cow	Veneto	<LOQ
9	Blue	Cow	Piedmont	<LOQ	51	Grated hard	Cow	Lombardy	<LOQ
10	Grated hard	Cow	Lombardy	<LOQ	52	Grated hard	Cow	Lombardy	<LOQ
11	Grated hard	Cow	Lombardy	<LOQ	53	Grated	Goat	Sardinia	<LOQ
12	Grated hard	Cow	Emilia Romagna	<LOQ	54	Grated hard	Ewe	Piedmont	<LOQ
13	Hard	Cow	Lombardy	<LOQ	55	Blue	Cow	Piedmont	<LOQ
14	Grated hard	Cow	Piedmont	<LOQ	56	Grated hard	Cow	Lombardy	<LOQ
15	Grated hard	Cow	Emilia Romagna	1.7	57	Grated hard	Cow	Lombardy	<LOQ
16	Grated hard	Cow	Lombardy	<LOQ	58	Blue	Cow	England	<LOQ
17	Grated hard	Cow	Lombardy	<LOQ	59	Grated hard	Cow	Veneto	<LOQ
18	Grated hard	Cow	Lombardy	<LOQ	60	Blue	Cow	France	0.0044
19	Grated hard	Cow	Emilia Romagna	<LOQ	61	Spicy	Cow	Veneto	<LOQ
20	Grated hard	Cow	Trentino	<LOQ	62	Blue	Cow	England	<LOQ
21	Grated hard	Cow	Lombardy	<LOQ	63	Grated hard	Cow	Puglia	7.5
22	Grated hard	Cow	Emilia Romagna	7.2	64	Grated hard	Cow	Lombardy	1.3
23	Grated hard	Cow	Emilia Romagna	<LOQ	65	Grated hard	Cow	Puglia	<LOQ
24	Grated hard	Cow	Lombardy	<LOQ	66	Grated hard	Cow	Umbria-Marche	<LOQ
25	Blue	Cow	Lombardy	<LOQ	67	Grated hard	Cow	Piedmont	<LOQ
26	Semi-hard	Ewe	Abruzzo	<LOQ	68	Grated hard	Cow	Lombardy	<LOQ
27	Soft	Cow	Trentino	<LOQ	69	Hard	Cow	Lombardy	<LOQ
28	Soft	Goat	n.a.	<LOQ	70	Soft-ripened	Cow	Lombardy	<LOQ
29	Spun paste	Cow	Lombardy	<LOQ	71	Hard	Cow	Emilia Romagna	<LOQ
30	Blue	Cow	Piedmont	<LOQ	72	Hard	Cow	Emilia Romagna	<LOQ
31	Blue	Cow	Lombardy	<LOQ	73	Hard	Cow	Emilia Romagna	<LOQ
32	Blue	Cow	Lombardy	<LOQ	74	Grated hard	Cow	Emilia Romagna	3.2
33	Blue	Cow	Piedmont	<LOQ	75	Grated hard	Cow	Emilia Romagna	5.0
34	Blue	Cow	Lombardy	<LOQ	76	Grated hard	Cow	Trentino	<LOQ
35	Blue	Cow	Piedmont	<LOQ	77	Grated hard	Cow	Emilia Romagna	<LOQ
36	Hard	Cow	Emilia Romagna	<LOQ	78	Grated hard	Cow	Lombardy	<LOQ
37	Grated hard	Cow	Lombardy	<LOQ	79	Grated hard	Cow	Lombardy	<LOQ
38	Grated hard	Cow	Emilia Romagna	<LOQ	80	Grated hard	Ewe	Tuscany	<LOQ
39	Grated hard	Cow	Emilia Romagna	<LOQ	81	Grated hard	Cow	Lombardy	<LOQ
40	Grated hard	Cow	Emilia Romagna	<LOQ	82	Grated hard	Cow	Lombardy	<LOQ
41	Blue	Goat	Piedmont	<LOQ	83	Grated hard	Cow	Emilia Romagna	22.4
42	Blue	Cow	Piedmont	<LOQ	84	Grated hard	Cow	Lombardy	<LOQ

^1^ Italian region or foreign state.

**Table 4 toxins-13-00540-t004:** Mass spectrometric parameters for the simultaneous determination of OTA and U-[^13^C_20_]-OTA (IS) using an electrospray interface (ESI) in positive ionisation mode.

Analyte	MW (g/mol)	Retention Time (min)	Precursor Ion (m/z)	Product Ions (m/z)	CE (eV)
Ochratoxin A	403.81	2.7	403.9	239.2 *	28
				357.8	12
U-[^13^C_20_]-ochratoxin A	423.67	2.7	423.8	377.2	15

* Quantification ion.

## Data Availability

Not applicable.

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
