# Peer review of "Occurrence of Ochratoxin A in Different Types of Cheese Offered for Sale in Italy"

_toxins, 2021, doi:10.3390/toxins13080540_

Round 1

Reviewer 1 Report

This paper reads well and is flowing. The data are clearly presented and easy to follow. I would advice to move in the introduction a detail specified in the conclusion (line 356). It is not clear if or that the grated cheese used comes from ready-to-use grated cheese packages until the conclusions. If this is the case, it should be moved to the introduction, or explain better the sources of the hard cheese used for testing. 

Also, while the focus of this article is on the presence of OTA in cheese, it's not offered enough data or information that address the effects in human health when this toxic element is ingested, beside mentioning as being a possible carcinogen element for humans (line 104), or what level of OTA can be considered high and cause illness in humans. I would suggest to offer some data that offer evidence about the effect in humans in terms of quantity of OTA in cheese, if there is any evidence available.

Reviewer 2 Report

The MS toxins-1298995 with the title of (Occurrence of ochratoxin A in different types of cheese offered for sale in Italy) investigated the presence of OTA in cheeses purchased in the metropolitan city of Bologna (Italy) and its surrounding area. Below are my comments:

The most concern issue was uncited sentences in the introduction or discussion sections.

For example, L39-43 who said this text??

Authors must cite such text! Same issue in L33-35,

L46-51 who said this?

L55-58 who said this?

 L63-68 who said this?

L138-140 who said this? Please cite all these sentences.

Introduction section: The authors wrote short paragraphs, then one long paragraph appeared in the front of my eyes (L61-97), please try to dive this long one into two paragraphs.

In the introduction section, I like the author to write something about food security and can link this with the COVID-19, they can write 2-5 sentences about this. Here is one related recent and published article about this issue, authors can cite it in the MS (Seleiman, M.F.; Selim, S.; Alhammad, B.A.; Alharbi, B.M.; Juliatti, F.C. Will novel coronavirus (COVID-19) pandemic impact agriculture, food security and animal sectors? Biosci. J. 202036, 1315–1326.).

Results section:

L162: Table 4??? How authors can cite Table 4 before Tables 1,2,3? Something is wrong here>>> Please correct!

Text in Figure 1 is small, authors can make it bigger and clear.

In the Tables, why the authors presented their data or means with SD (Standard deviation)? Why they did not use SE (Standard error) instead of SD? SE is much common and useable than SD!

Discussion section: Authors must cite the sentences that are not from their results or not their own.

Add [66] after Biancardi et al. in L271, and remove [66] from L273.

In conclusion, I prefer the authors avoid using citations such as Biancardi et al.

In material and methods section: authors should cite the different methods that were used in the different analysis, I am sure that some of methods are not their own.

Round 2

Reviewer 2 Report

The MS has been improved. Here is only one comment

L37: Cite also ref number 1 with ref number 2

Author Response

As requested we have cited also ref number 1 with ref number 2.